# Ditto: Scaling Instruction-Based Video Editing with a High-Quality Synthetic Dataset

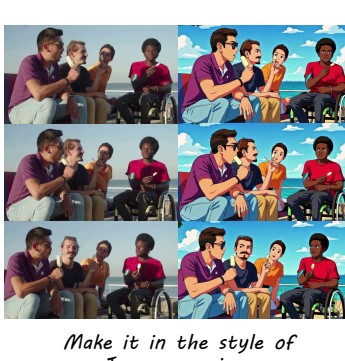
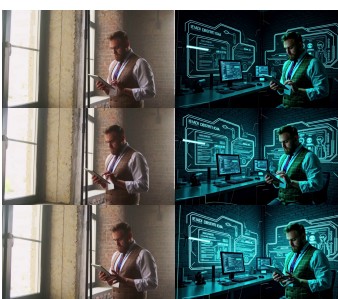
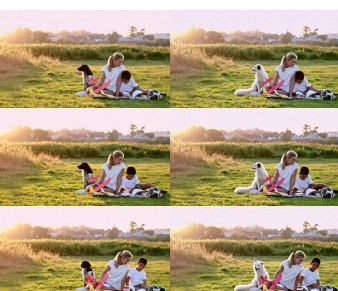

*Make it in the style of Japanese anime.*

*Reimagine the industrial setting in a cybernetic brain lab where the tablet is a neural interface.*

*Replace the black dog with a white fox sitting calmly beside them.*

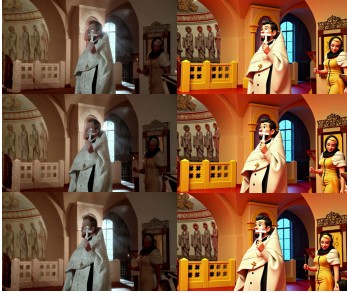
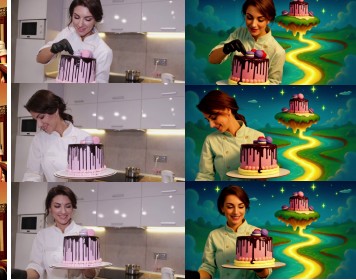
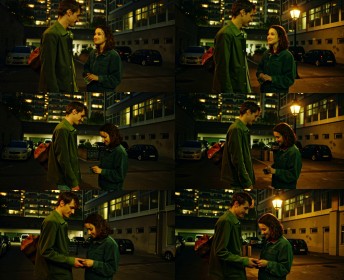

*Imitate the look of the 3D Chibi style.*

*Transform the scene into a whimsical pastry dream world where the cake is a floating island.*

*Add a glowing vintage streetlamp casting a warm yellow hue on the pavement near the couple.*

Figure 1: Our proposed synthetic data generation pipeline can automatically produce high-quality and highly diverse video editing data, encompassing both global and local editing tasks. We highly recommend the readers to see the supplementary video samples.

## Abstract

Instruction-based video editing promises to democratize content creation, yet its progress is severely hampered by the scarcity of large-scale, high-quality training data. We introduce *Ditto*, a holistic framework designed to tackle this fundamental challenge. At its heart, Ditto features a novel data generation pipeline that fuses the creative diversity of a leading image editor with an in-context video generator, overcoming the limited scope of existing models. To make this process viable, our framework resolves the prohibitive cost-quality trade-off by employing an efficient, distilled model architecture augmented by a temporal enhancer, which simultaneously reduces computational overhead and improves temporal coherence. Finally, to achieve full scalability, this entire pipeline is driven by an intelligent agent that crafts diverse instructions and rigorously filters the output, ensuring quality control at scale. Using this framework, we invested over 12,000 GPU-days to build *Ditto-1M*, a new dataset of one million high-fidelity video editing examples. We trained our model, *Editto*, on Ditto-1M with a curriculum learning strategy. The results demonstrate superior instruction-following ability and establish a new state-of-the-art in instruction-based video editing. We will release our dataset, models, and code to accelerate research in this field.

# 1 INTRODUCTION

Recently, the field of visual generative models has witnessed a remarkable divergence: while instruction-based *image* editing has achieved unprecedented levels of precision and user-friendliness with models like InstructPix2Pix (Brooks et al., 2023), FLUX.1 Kontext (Batifol et al., 2025), Qwen-Image (Wu et al., 2025a), and Gemini 2.5 Flash Image (Nano-Banana) (Google, 2025), its *video* counterpart has lagged significantly behind. This growing capabilities gap stems from the inherent complexities of the temporal dimension. Editing a video requires not only modifying content but also ensuring these changes are propagated coherently across frames—a challenge that has proven formidable. The primary obstacle impeding progress is a well-understood but unsolved problem: the profound scarcity of large-scale, high-quality, and diverse paired data for training end-to-end video editing models (Zhang et al., 2025; Yu et al., 2025).

Existing works have attempted to address this data scarcity challenge through various synthetic data generation strategies. Earlier approaches either relied on computationally prohibitive per-video optimization methods (Qin et al., 2024) or adopted training-free image-to-video propagation techniques (Yu et al., 2025; Wu et al., 2025b). However, these pipelines suffer from a persistent trade-off: they sacrifice editing diversity, temporal consistency, and visual quality for scalability, or vice-versa. A scalable, cost-efficient data pipeline that generate high-fidelity results remains an open challenge.

To address these shortcomings, we introduce **Ditto**[1], a scalable and cost-efficient data synthesis pipeline architected to systematically dismantle these trade-offs. Our approach first tackles the challenge of editing fidelity and diversity. Capitalizing on the advanced maturity of instruction-based image editors, the pipeline generates a high-quality edited reference frame that acts as a strong visual prior. This anchor frame then guides an in-context video generator (Jiang et al., 2025) to synthesize a temporally coherent video that faithfully matches the edit, overcoming the quality limitations of previous methods. Second, to resolve the critical efficiency-coherence trade-off where high-fidelity generation is prohibitively expensive, our pipeline integrates a distilled video model with a temporal enhancer. This innovative combination reduces computational costs to just 20% of the original while preserving temporal stability and avoiding visual artifacts. Finally, to achieve true scalability and eliminate the bottleneck of manual curation, we deploy an autonomous Vision-Language Model (VLM) agent. This agent carries dual responsibilities: programmatically generating diverse instructions for both local and global edits, and serving as a flaw-detection mechanism to automatically filter out low-quality or failed video pairs, ensuring the final dataset's integrity.

We invested over *12,000* GPU-days using this pipeline to construct **Ditto-1M**, a new large-scale dataset comprising over *one million* source-instruction-edited video triplets, as demonstrated in Fig. 1. The dataset is meticulously structured to cover a wide spectrum of editing tasks and is curated via our VLM agent to ensure instruction consistency and high aesthetic quality.

With the proposed dataset, we train our final editing model, **Editto**. To bridge the gap between our visually-guided data synthesis and the goal of purely instruction-driven inference, we propose a modality curriculum learning strategy (Bengio et al., 2009). Our curriculum begins by providing the model with both the text instruction and the edited reference image as a "scaffold." As training progresses, we gradually anneal the visual guidance, compelling the model to learn the more difficult, abstract mapping from text instruction alone.

Our contributions are as follows:

- A novel, scalable synthesis pipeline, Ditto, that efficiently generates high-fidelity and temporally coherent video editing data.

- The Ditto-1M Dataset, a million-scale, open-source collection of instruction-video pairs to facilitate community research.

- A state-of-the-art editing model, trained on Ditto-1M, that demonstrates superior performance on established benchmarks.

- A modality curriculum learning strategy that effectively enables a visually-conditioned model to perform language-driven editing.

---

[1]The name "Ditto" is chosen to reflect the model's core function: making the output video a faithful reflection, or "ditto," of the user's textual instruction.

## 2 RELATED WORK

### 2.1 INSTRUCTION-BASED IMAGE EDITING

Visual generative models have advanced rapidly (Wan et al., 2025; Ma et al., 2025; Rombach et al., 2022; Saharia et al., 2022; Blattmann et al., 2023; Guo et al., 2023; Kong et al., 2024; Song & Dhariwal, 2023; Lin et al., 2025; He et al., 2024). Instruction-based image editing has also rapidly evolved, moving beyond simple text-to-image generation to enable nuanced, user-guided modifications. Early and influential methods like InstructPix2Pix (Brooks et al., 2023) demonstrated the feasibility of fine-tuning diffusion models on generated datasets of image triplets (source image, instruction, edited image) to perform edits. This was achieved by ingeniously combining a large language model (GPT-3) to generate textual edit instructions and a text-to-image model (Stable Diffusion) to synthesize the corresponding image pairs, creating a large-scale training corpus without manual annotation. More recent advancements, particularly with the advent of powerful models like FLUX.1 Kontext (Batifol et al., 2025), Qwen-Image (Wu et al., 2025a), and Gemini 2.5 Flash Image (Nano-Banana) (Google, 2025), have unlocked even more sophisticated capabilities. These models can process both text and reference images as inputs, enabling targeted local edits, robust character consistency across multiple turns, and complex scene transformations within a unified architecture, often without requiring fine-tuning. Our work builds upon this progress by integrating a state-of-the-art instruction-based image editor as a critical component in our video data synthesis pipeline, using it to manipulate keyframes that guide the subsequent video-level edit.

### 2.2 INSTRUCTION-BASED VIDEO EDITING

Video editing has gain remarkable progress (Yang et al., 2025; Ceylan et al., 2023; Liu et al., 2024; Chai et al., 2023) with the development of the base generative models. Extending instruction-based editing to video requires maintaining temporal consistency and preserving background content. Current approaches fall into two main categories:

**Inversion-based Methods.** These methods avoid paired video-text-edit data but are computationally intensive. Tune-A-Video (Wu et al., 2023) fine-tunes a text-to-image model on a single video, enabling personalized edits but lacking scalability. Zero-shot techniques like TokenFlow (Geyer et al., 2024) and FateZero (Qi et al., 2023) use DDIM inversion and feature propagation to enforce the consistency of the edited video. However, their quality relies on inversion fidelity and often struggles with complex motion or occlusions.

**Feed-forward Methods.** These end-to-end models aim to overcome inversion-based limitations but face the fundamental challenge of data scarcity. The development of feed-forward approaches is tightly coupled with the creation of synthetic datasets, as large-scale human-annotated video edit data is notoriously scarce. Early works (Qin et al., 2024; Zhang et al., 2024) attempted to bypass this bottleneck by synthesizing data using one-shot tuning methods like Tune-A-Video (Wu et al., 2023) or CoDeF (Ouyang et al., 2024). However, this process was computationally expensive and ultimately limited the scale and quality of the resulting dataset. More recent methods have adopted a "lift and propagate" paradigm for more scalable data generation. VEGGIE (Yu et al., 2025) and InsViE (Wu et al., 2025b), for instance, leverage high-quality image editing datasets by applying an image editor to a keyframe and then using an image-to-video model to generate or propagate the edit across the entire video clip. While this strategy improves scalability, the temporal coherence and visual quality of the final output are fundamentally capped by the image-to-video propagation model, which often introduces unnatural motion or identity inconsistencies. These methods also typically rely on training-free inversion or attention control techniques during data synthesis, which may limit editing flexibility. In contrast, our approach introduces a data synthesis pipeline centered around an in-context video generator that conditions on both a reference edited frame and a depth-derived motion representation. This allows for more direct and high-quality video synthesis.

## 3 DITTO-1M

Our methodology begins with the construction of a large-scale, high-quality dataset. We designed a novel, scalable data generation pipeline to synthesize over one million instruction-video triplets,

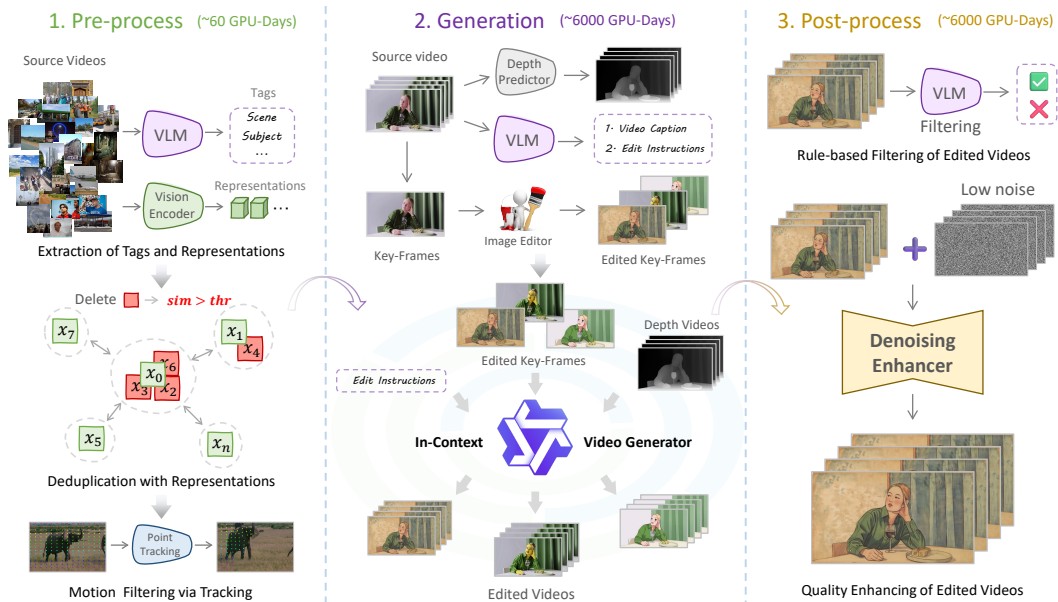

Figure 2: Our scalable data synthesis pipeline. (1) Pre-processing: A diverse video pool is curated via automated deduplication and motion filtering. (2) The core engine synthesizes video triplets, conditioning an in-context generator on automated instructions, appearance context from edited key-frames, and structural context from depth maps. (3) Post-processing: Final visual quality is guaranteed by a VLM-based filter and a denoising enhancer.

as in Fig. 2. The architecture of this pipeline was specifically engineered to address four critical challenges inherent to existing data synthesis approaches:

1. **Overcoming Limited Editing Diversity and Fidelity**. Current data pipelines of instruction-based video editing often rely on training-free inversion techniques (Qin et al., 2024; Zhang et al., 2025), which tend to yield synthetic data of limited quality. To address this, we propose to leverage an in-context video generator to produce high-quality editing samples with visual contexts. Capitalizing on the more advanced development of image-based editing models, we incorporate strong priors from these image editors to serve context and guide the video generation for better editing quality. This is combined with depth-guided video context to ensure spatiotemporal coherence, significantly improving the diversity and fidelity of generated edits.

2. **Resolving the Efficiency-Quality Trade-off**. A major technical hurdle is the trade-off between generation cost and data quality. Current high-fidelity methods are prohibitively expensive (e.g., 50 GPU-minutes per sample on a single GPU), while faster, distilled models often introduce artifacts like temporal flickering. Our pipeline is designed with a cost-aware workflow that significantly reduces computational overhead without compromising the temporal coherence of the videos.

3. **Automating Instruction Generation and Quality Control**. To achieve true scalability, manual creation of instructions and verification of outputs is infeasible. Our pipeline integrates an automated agent with two primary responsibilities: (a) programmatically generating diverse and meaningful instructions for both local and global edits, and (b) serving as a flaw-detection mechanism to automatically filter out generated pairs that are of low quality or fail to follow the instructions.

4. **Ensuring High Aesthetic and Motion Quality**. Unlike general-purpose video datasets (e.g., Panda-70M), which are not optimized for editing tasks, our pipeline prioritizes the generation of content with high aesthetic value and natural motion dynamics. This focus ensures the resulting dataset, and the models trained upon it, are well-aligned with real-world usage scenarios where visual appeal is paramount.

The following sections will detail the architecture of our data generation pipeline, explaining how each component systematically addresses these challenges.

Figure 3: Source categories.

| Datasets | Amount | Resolution | Frames | FPS | Real | Filter |
|---|---|---|---|---|---|---|
| InstructVid2Vid (Qin et al., 2024) | N/A | N/A | N/A | N/A | ✓ | ✗ |
| EffiVED Zhang et al. (2024) | 155k | 512×512 | 8 | N/A | ✓ | ✗ |
| InsV2V (Cheng et al., 2024) | 404k | 256×256 | 16 | N/A | ✗ | ✗ |
| InsViE (Wu et al., 2025b) | 1M | 1024×576 | 25 | 7 | ✓ | ✓ |
| **Ours** | 1M | 1280×720 | 101 | 20 | ✓ | ✓ |

Table 1: Comparisons with prior instruction-based datasets.

## 3.1 SOURCE VIDEO FILTERING

The Ditto-1M dataset is built exclusively from high-resolution videos sourced from Pexels (Pexels, 2025), a platform for professional-grade footage under the *Pexels License*. Unlike datasets derived from uncurated web scrapes, this strategy provides a foundation of superior aesthetic and technical quality, suitable for video editing tasks. We also first apply a rigorous filtering and pre-processing protocol. This protocol examines videos in the following aspects:

**Near-Duplicate Removal:** To prevent dataset redundancy and ensure broad content diversity, we implement a rigorous deduplication process. We employ a powerful visual encoder (Oquab et al., 2023) to extract compact feature representations for each video. Pairwise similarity between these feature vectors is then computed. Videos exceeding a pre-defined similarity threshold are systematically filtered out, guaranteeing the uniqueness of each source video in our collection.

**Motion Scale:** Videos that contain little or no motion over time—such as fixed-camera surveillance footage, still nature scenes, or unmoving interior shots—are considered less valuable for video editing tasks because they lack dynamic visual changes. To automatically identify such low-dynamic content, we employ a tracking-based method that analyzes frame-to-frame motion across the video sequence. Specifically, for each video, we first sample points on a grid layout and then use Co-Tracker3 (Karaev et al., 2024) to track these points, obtaining their trajectories. We then compute the average of the cumulative displacements of all tracked points over the entire video as the motion score of the video. By setting a threshold, we filter out videos with low motion scores, effectively removing those with negligible temporal variation. Videos that pass this filtering stage are then standardized. Each video is resized to a uniform resolution and its frame rate is converted to 20 FPS. This standardization simplifies the training process and ensures consistency across the entire dataset.

## 3.2 INSTRUCTION GENERATION

For each filtered source video $V_s$, we generate a set of corresponding editing instructions $p$. We employ a powerful VLM (Bai et al., 2025) for this task with a two-step prompting strategy. First, we prompt the VLM to generate a dense caption $c$ that describes the video's content, subjects, and scenery:

$$c = \text{VLM}(V_s, p_{caption}). \tag{1}$$

This caption serves as a semantic anchor. Next, we feed both the video $V_s$ and its caption $c$ back into the VLM, prompting it to devise a creative and plausible editing instruction $p$:

$$p = \text{VLM}(V_s, c, p_{instruct}). \tag{2}$$

This conditioned approach ensures that instructions are contextually grounded in the video's content, yielding a diverse set of commands ranging from global style transformations to specific, localized object modifications.

## 3.3 VISUAL CONTEXT PREPARATION

Our generation process is heavily guided by a rich visual context, which consists of two key components: an edited reference frame that specifies the target appearance, and a depth video that enforces spatiotemporal consistency.

**Key-Frame Editing for Appearance Guidance.** We first select a key-frame $f_k$ from the source video $V_s$ as the anchor for the editing. This frame is then edited by the instruction-guided image editor $\mathcal{E}_{\text{img}}$ (Wu et al., 2025a), using the instruction p generated in the previous step:

$$f'_k = \mathcal{E}_{\text{img}}(f_k, p). \tag{3}$$

This resulting frame $f'_k$, serves as the visual prototype for the edit, defining the final appearance including style and textures.

**Depth Video Prediction for Spatiotemporal Structure.** To preserve the geometric structure and motion dynamics of the original scene, we extract a dense depth video $V_d$ from $V_s$ with a video depth predictor $\mathcal{D}$ (Chen et al., 2025). The predicted depth video acts as a dynamic structural scaffold, providing an explicit, frame-by-frame guide for the structure and geometry of the scene during the video generation.

### 3.4 IN-CONTEXT VIDEO GENERATION

With the editing instruction $p$ and the multi-modal visual context $f'_k$ and $V_d$ prepared, we synthesize the edited video $V_e$ with the in-context video generator (Jiang et al., 2025), which is denoted as $\mathcal{G}$. VACE is a feed-forward video generative model designed to condition its generation on rich visual prompts such as images, masks, and videos by learning a context branch beyond the base generative model (Wan et al., 2025). In our design, we adopt $\mathcal{G}$ to synthesize the edited video by taking the textual prompt p as a high-level semantic guide, the edited key-frame $f'_k$ as the primary appearance condition, and the depth video $V_d$ as a strict spatiotemporal constraint. This generation process is formulated as:

$$V_e = \mathcal{G}(V_d, f'_k, p). \tag{4}$$

By integrating these three modalities with the attention mechanism, VACE can faithfully propagate the edit defined in $f'_k$ across the entire sequence, adhering to the motion and structure laid out by $V_d$, while ensuring the result is semantically aligned with the instruction $p$. Our pipeline achieves high-quality and coherent video edits without costly per-video optimization. Please refer to Appendix B for additional analysis of the data generator. To facilitate scalable synthetic data generation and further reduce the computational burden, we employ model quantization and knowledge distillation techniques (Yin et al., 2025). We apply post-training quantization to reduce the model's memory footprint and inference cost with minimal impact on output quality. Furthermore, we adopt the generative video model (Yin et al., 2025) distilled from the teacher model, preserving editing fidelity while significantly accelerating the generation process with few-step inference. This optimized pipeline is crucial for producing large-scale video editing data efficiently.

### 3.5 EDITED VIDEO CURATION AND ENHANCING

To guarantee the highest quality, the generated triplets $(V_s, p, V_e)$ undergo a final two-stage curation and refinement including VLM filtering and denoiser enhancing.

**VLM-Based Curation.** We first use a VLM (Bai et al., 2025) as an automated judge to perform rejection sampling. Each triplet is evaluated against two criteria: (1) Instruction Fidelity: whether the edit in $V_e$ accurately reflects the prompt $p$. (2) Fidelity: whether $V_e$ preserves the semantic and motion from $V_s$. (3) Visual quality: whether the videos are visual appealing without significant distortion or artifacts. (4) Safety & Appropriateness: whether the content has unsafe or inappropriate material, such as pornography, violence, or horror, ensuring the dataset is ethically compliant and suitable. Triplets that fail to meet our quality thresholds on these criteria are discarded.

**Quality Enhancement via Denoising.** The curated edited videos are then enhanced using the state-of-the-art open-source Text-to-Video (T2V) model, Wan2.2 (Wan et al., 2025). Unlike post-processing in prior work that performs simple upscaling, our objective is to achieve perceptual refinement without introducing semantic deviations from edited content of $V_e$. This requirement aligns perfectly with the specialized design of Wan2.2's Mixture-of-Experts (MoE) architecture, which employs a coarse denoiser for structural and semantic formation under high noise, and a fine denoiser specialized in later-stage refinement under low noise. We specifically leverage the fine denoiser for a short, 4-step reverse process. For each video $V_e$, we first add a small amount of Gaussian noise. The fine denoiser then inverts this process utilizing its expert prior to remove subtle artifacts and enhance textural details precisely because it is optimized for making minimal, semantic-preserving adjustments to nearly-complete videos. This yields a high-quality output with improved resolution and visual fidelity that remains strictly semantically consistent with our initial edit.

Figure 4: Model training pipeline. We train the context blocks based on the in-context video generator with curriculum learning by gradually annealing and eventually dropping the reference frame.

Table 2: Quantitative comparisons with prior arts. The best results are **bolded**.

| Method | Automatic Metric | | | Human Evaluation | | |
|---|---|---|---|---|---|---|
| | CLIP-T ↑ | CLIP-F ↑ | VLM ↑ | Edit-Acc ↑ | Temp-Con ↑ | Overall ↑ |
| TokenFlow (Geyer et al., 2024) | 23.63 | 98.43 | 7.10 | 1.70 | 1.97 | 1.70 |
| InsV2V (Cheng et al., 2024) | 22.49 | 97.99 | 6.55 | 2.17 | 1.96 | 2.07 |
| InsViE (Wu et al., 2025b) | 23.56 | 98.78 | 7.35 | 2.28 | 2.30 | 2.36 |
| **Ours** | **25.54** | **99.03** | **8.10** | **3.85** | **3.76** | **3.86** |

## 3.6 DETAILS OF DITTO-1M

As illustrated in Fig. 3, we collected a total of over 200k source videos, approximately half of which feature human activities. After undergoing a filtering process, these videos were edited using editing instructions generated by a VLM, followed by an additional round of filtering. This pipeline ultimately yielded approximately 1M edited videos. Among these, about 700k video triplets involve global editing (including changes to style, environment, etc.), while roughly 300k pertain to local editing (encompassing object replacing, adding, and removal). As presented in Table 1, the final enhanced videos have a resolution of 1280x720, each comprising 101 frames at 20 FPS. The visual quality of the final samples significantly surpasses that of previous datasets - we strongly recommend reviewing the video samples provided in the supplementary materials.

## 4 MODEL TRAINING VIA MODALITY CURRICULUM LEARNING

We select the in-context video generator VACE (Jiang et al., 2025) as our backbone, inspired by its strong prior for generating videos that are spatially and structurally aligned with a source video. VACE's original capability is to condition its generation on two visual contexts (and prompts): a source video and a reference image. Our goal is to repurpose this powerful visual generator into a proficient editor that operates on abstract textual instructions. However, directly fine-tuning the model to bridge the vast semantic gap from visual to textual conditioning is prone to instability. We therefore adapt its architecture, as shown in Fig. 4. It consists of a Context Branch for extracting spatiotemporal features from the source video and reference frame, and a DiT-based (Peebles & Xie, 2023) Main Branch that synthesizes the edited video under the joint guidance of the visual context and the new textual embeddings from the instruction.

To ease the training difficulty and stably bridge this modality gap, we introduce a modality curriculum learning (MCL) strategy. The core idea is to leverage the model's inherent ability to process the reference image context as a temporary aid. In the initial training phase, we provide the edited reference frame as a strong visual "scaffold" alongside the new text instruction. As training progresses, we gradually anneal the probability of providing this visual scaffold, eventually dropping it entirely. This process compels the model to shift its dependency from the concrete visual target it already understands to the more abstract textual instruction, transforming it into a purely instruction-based video editing model. We train the model using the flow matching (Lipman et al., 2022) objective:

$$\mathcal{L} = \mathbb{E}_{t,\mathbf{z}_0,\mathbf{c}} \|\mathbf{v}_t(\mathbf{z}_t, t, \mathbf{c}) - (\mathbf{z}_0 - \mathbf{z}_t)\|^2, \quad (5)$$

where $\mathbf{z_0}$ is the clean latent encoded from the target edited video, $\mathbf{z_t}$ is its noised version at timestep $t$, $\mathbf{c}$ represents the conditioning from text and visual contexts, and $\mathbf{v}_t$ is the model's predicted vector field pointing from $\mathbf{z_t}$ to $\mathbf{z_0}$.

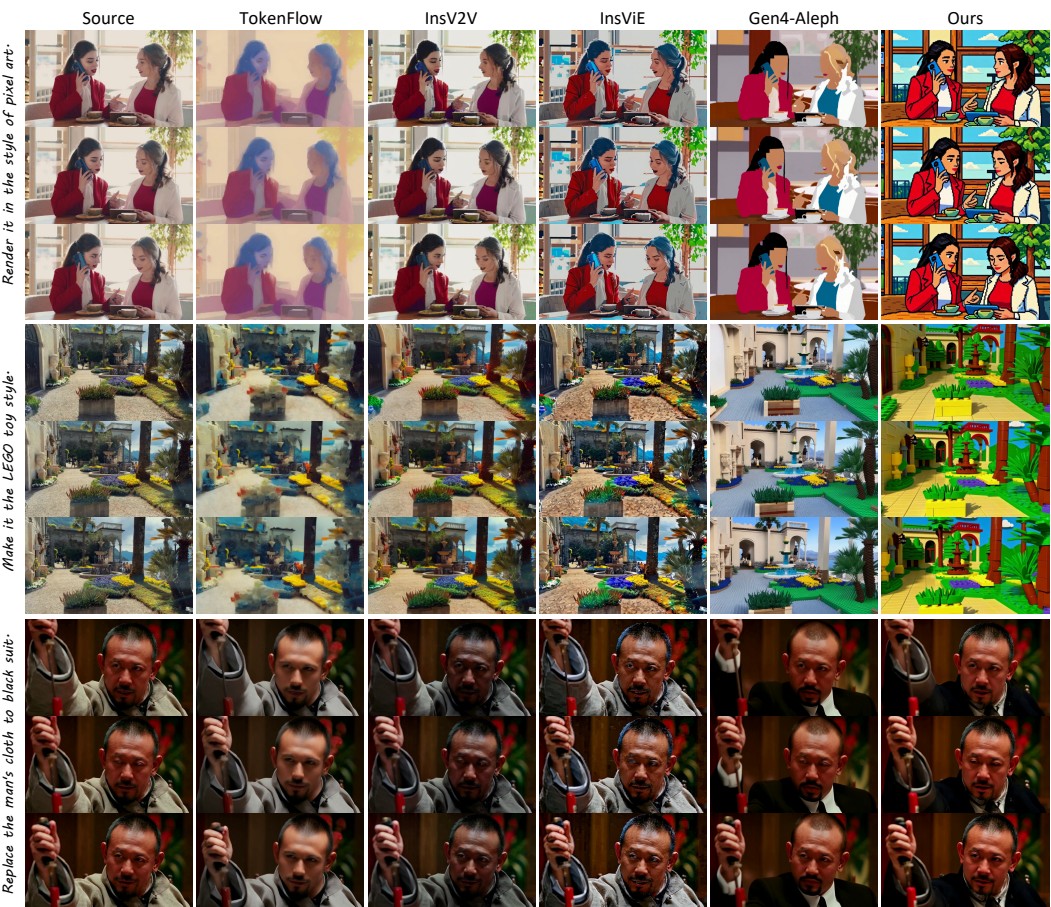

Figure 5: Qualitative comparisons with prior arts TokenFlow (Geyer et al., 2024), InsV2V (Cheng et al., 2024), InsViE (Wu et al., 2025b) and Gen4-Aleph (Runway, 2025).

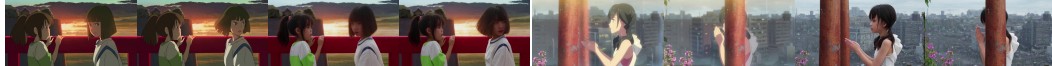

Figure 6: Our data and learned model enable the translation from synthetic videos to the real domain.

## 5 EXPERIMENTS

### 5.1 EXPERIMENTAL SETTINGS

Our model is built upon the pre-trained in-context video generator (Jiang et al., 2025; Wan et al., 2025) and is fine-tuned on our newly proposed large-scale dataset, which comprises over one million high-quality video triplets. To maintain the strong generative prior of the base model and ensure training efficiency, we freeze the majority of the pre-trained model's parameters, and only fine-tune the linear projection layers of context blocks. The model is trained for approximately 16,000 steps using the AdamW optimizer (Loshchilov & Hutter, 2017) with a constant learning rate of 1e-4 on a cluster of 64 NVIDIA H-series GPUs. We employ our modality curriculum learning strategy, where the initial 5,000 steps serve as a curriculum warm-up phase.

### 5.2 EXPERIMENTAL RESULTS

**Quantitative Comparison.** We perform quantitative comparisons using automatic metrics and a user study, summarized in Table 2. For automatic evaluation, we employ three metrics: CLIP-T measures the CLIP text-video similarity to assess how well the edit follows the instruction; CLIP-F calculates the average inter-frame CLIP similarity to gauge temporal consistency; and a VLM score provides a holistic assessment of edit effectiveness, semantic preservation, and overall aesthetic quality. A user study based on 1,000 ratings also rates instruction-following (Edit-Acc), temporal

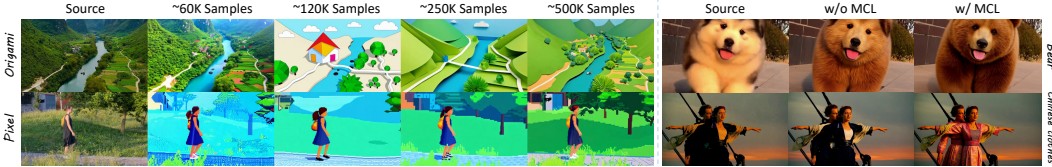

Figure 7: Unlike the original data generator, which fails to handle newly emerging information *beyond key frames*, our model - trained with filtering and scaling techniques - outperforms it.

Figure 8: Ablation studies on training data scale and modality curriculum learning (MCL).

consistency (Temp-Con), and overall quality (Overall). As shown, our method significantly outperforms all baselines across metrics, achieving the highest automatic scores and a strong preference in human evaluations, which confirms its superior instruction adherence, temporal smoothness, and visual quality. Please refer to Appendix C for additional details of the user study.

**Qualitative Comparison.** As shown in Fig. 5, our method consistently produces visually superior results that better adhere to edit instructions compared to prior arts. For complex stylizations, our model generates temporally coherent videos that accurately match the target style, while competitors often yield blurry or inconsistent results. For local attribute changes (e.g., "black suit"), our method precisely edits the target object while preserving identity and background details, a task where Gen4-Aleph slightly change the man's identity and other methods largely fail. We strongly recommend reviewing the video samples provided on the supplementary site for a better understanding.

**Additional Results.** We showcase the synthetic-to-real (sim2real) capability in Fig. 6 benefited from our data by training the model to map the stylized videos in our dataset back to their original, real-world source videos. This successful transfer highlights the rich and photorealistic information contained within our dataset, demonstrating its utility beyond standard editing tasks. Also, our final trained model substantially outperforms the raw data generator from our pipeline, demonstrating superior handling of newly emerged content as in Fig. 7. This superiority stems from our scaled training regimen, including the curriculum learning and exposure to the filtered, high-quality data.

## 5.3 ABLATION STUDIES

We conduct ablation studies to validate the key components of our framework, with results presented in Fig. 8. We find that our model's performance scales effectively with the training data - as the number of samples increases, both the quality of the stylistic edits and the fidelity to the original video's content and motion improve significantly, confirming the value of our large-scale dataset. Furthermore, we ablate our modality curriculum learning (MCL) strategy and find that, without MCL, the model often struggles to interpret the instruction's full semantic intent. Therefore it is crucial for bridging the modality gap and learning to follow instructions.

## 6 CONCLUSION

We have presented Ditto, a scalable framework that significantly advances instruction-based video editing by systematically addressing the core challenge of data scarcity through a new paradigm for large-scale data synthesis. Our synthetic data generation pipeline overcomes the fidelity-diversity and efficiency-coherence trade-offs plaguing prior methods by leveraging strong image-editing priors, a distilled in-context video generator with a temporal enhancer, and autonomous VLM-based quality control. This enables the creation of the large-scale, high-quality Ditto-1M dataset. The proposed modality curriculum learning strategy further ensures our model achieves state-of-the-art performance by effectively transitioning from visual-textual conditioning to purely instruction-driven inference. Extensive results reveal the effectiveness of the proposed method. For *large language model usage*, we used it solely to polish the writing and improve the fluency of this document.

ETHICS STATEMENT

This work advances instruction-driven video editing, with potential applications in creative content production and video post-processing. To mitigate the misuse risks, all experiments are conducted using publicly available video datasets. No sensitive data is used in training or evaluation. Our model is intended for responsible research and creative applications only. We strongly advocate for transparent labeling of AI-generated video content and support the development of detection mechanisms and watermarking techniques to prevent misuse.

REPRODUCIBILITY STATEMENT

To facilitate reproducibility, we will open-source the dataset and model upon publication. We believe that releasing these resources will accelerate future research in instruction-based video editing.

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

## A APPENDIX OVERVIEW

In this appendix, we provide additional details and results to supplement our main paper. In Appendix B, we present an analysis of the in-context data generator. We showcase the interface used for our human evaluations in Appendix C. Finally, Appendix D contains more qualitative results of our dataset and model, and a word cloud visualizing the distribution of instructions in our dataset. We strongly recommend reviewing the video samples in **"index.html"** in Supplementary Materials for a better understanding.

## B JUSTIFYING THE DESIGN OF DATA GENERATION PIPELINE

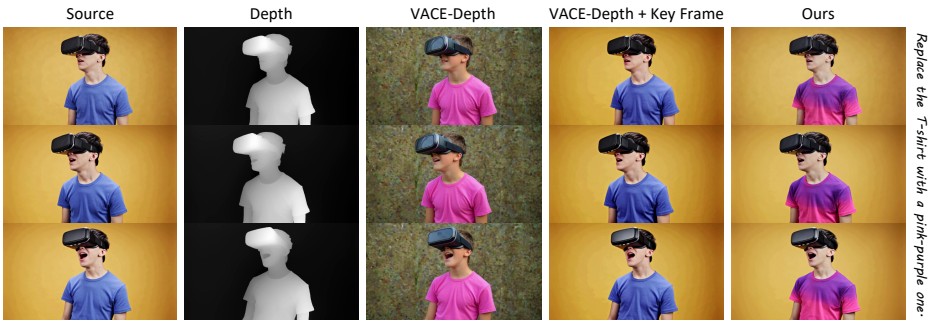

Figure 9: Results of various settings for data generation.

We provide an analysis of the videos synthesized by the in-context video generator, VACE, to justify the design of our data pipeline. As in Fig. 9, we first observe that using only depth maps to guide the generator results in a significant loss of content from the source video, leading to poor fidelity. Conversely, conditioning the generator on a keyframe from the original source video alongside the editing instruction fails to produce the desired edit - the output remains almost identical to the source. These findings reveal that while the base generator excels at motion transfer, its inherent instruction-following capability for editing tasks is limited. Based on this analysis, we validate our proposed approach: using a keyframe modified by an advanced image editor, in conjunction with depth guidance as the context. This method achieves the optimal balance of instruction adherence, temporal consistency, and source fidelity for our data synthesis.

## C DEMONSTRATION OF USER STUDY INTERFACE

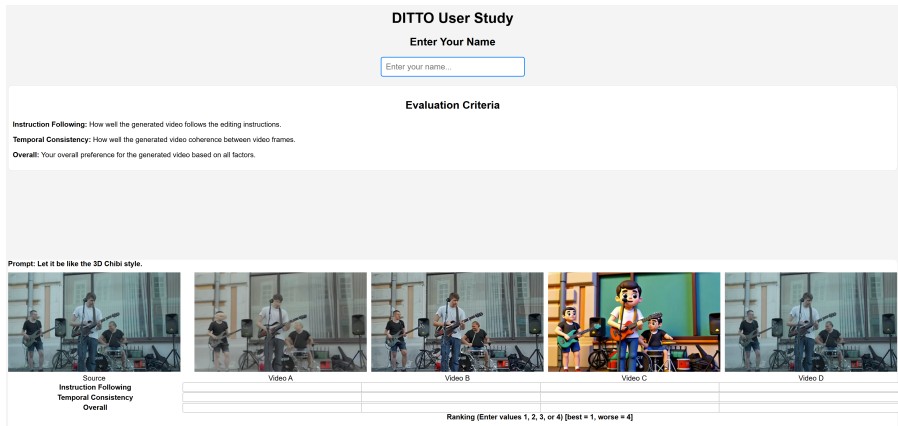

Figure 10: The interface of the user study.

To collect human preference data, we designed a user-friendly evaluation interface, as shown in Fig. 10. For each source video and text prompt, we presented participants with the edited results

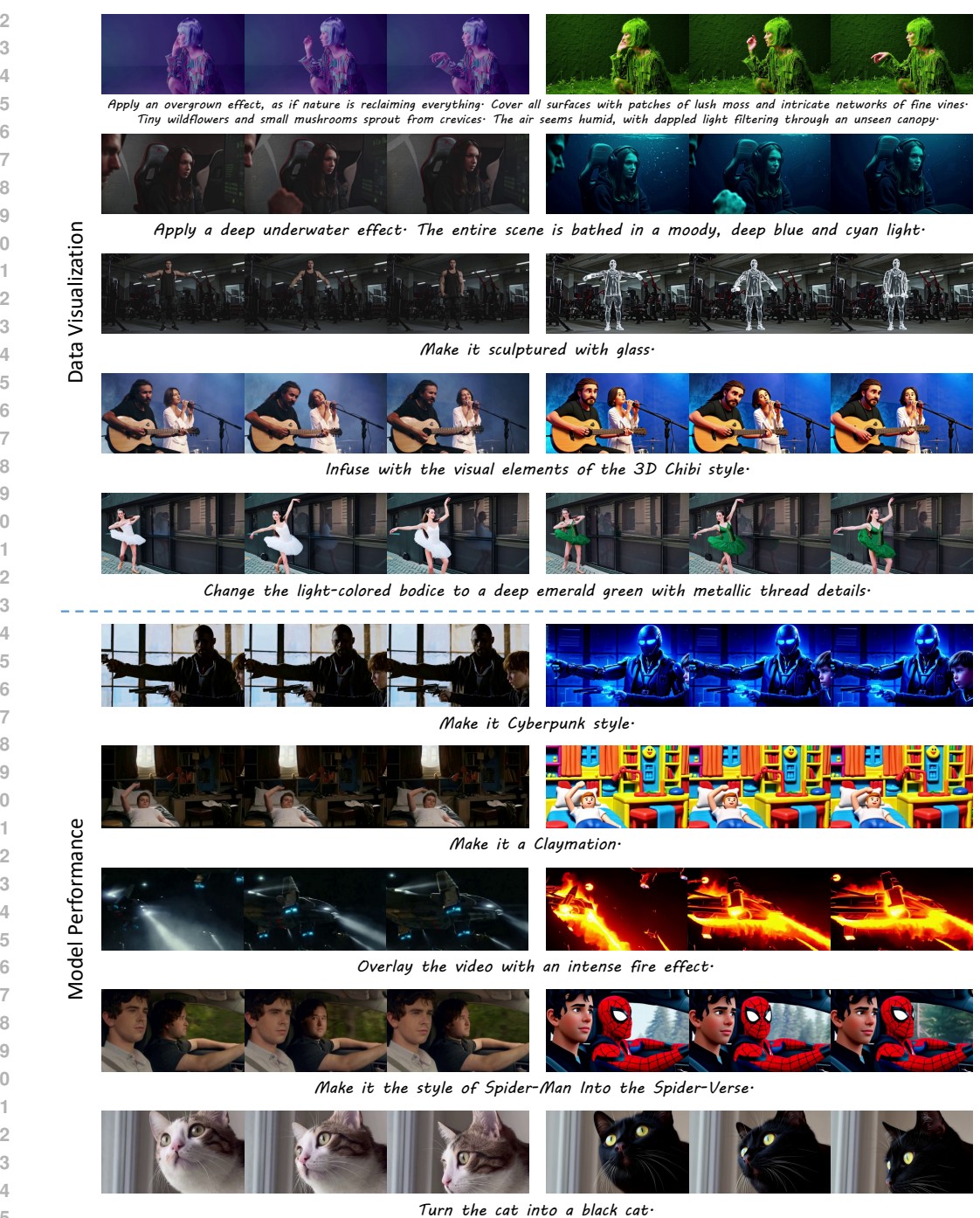

Figure 11: Additional visualization of data from the proposed dataset and model outputs. Please view the site in supplementary materials for additional video samples.

from different methods in a randomized order. They were then asked to rank the videos from best (1) to worst (4) based on three criteria: Instruction Following, Temporal Consistency, and Overall preference. The final scores are calculated based on the ranking.

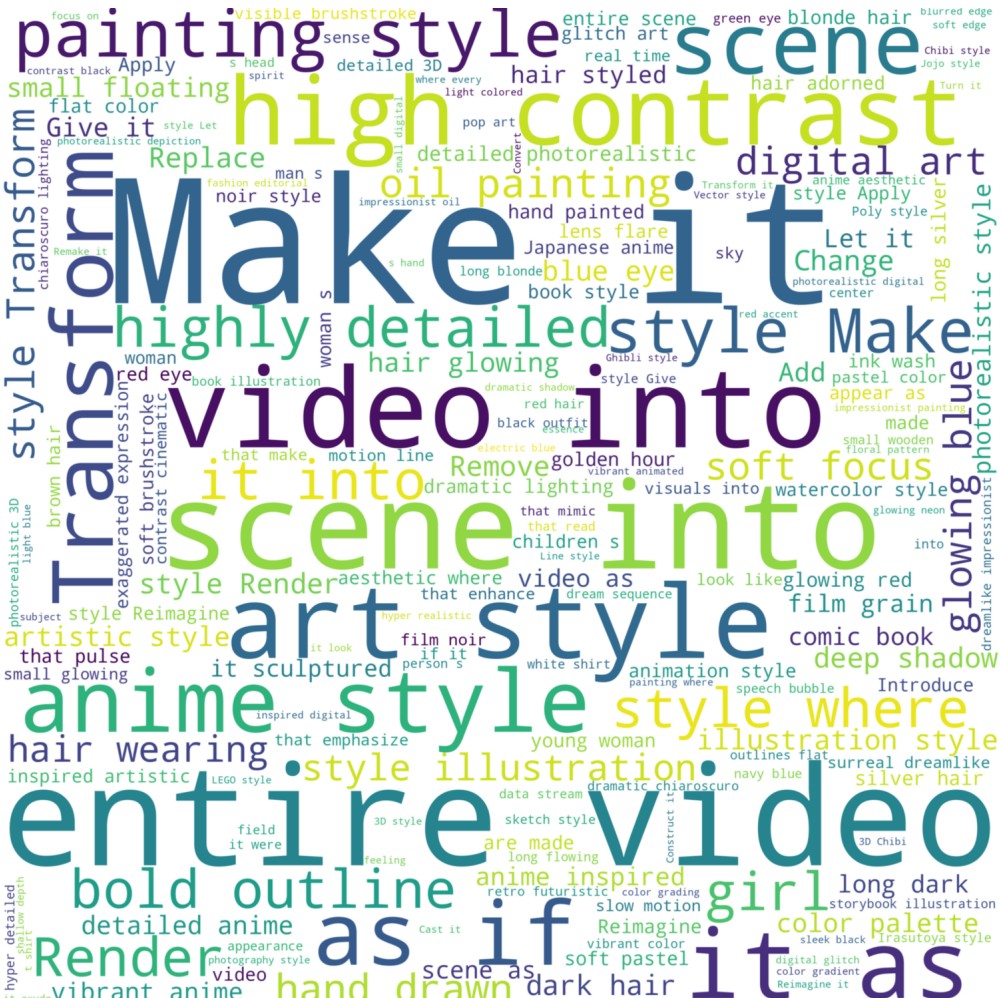

Figure 12: The word cloud of editing instructions.

## D  ADDITIONAL RESULTS OF DATASET AND MODEL

We present additional qualitative results to further demonstrate our dataset and model's performance across a wide range of editing instructions in Fig. 11. We also include a word cloud in Fig. 12 that illustrates the diversity and distribution of the editing prompts within the dataset, highlighting its comprehensive coverage. Additional video samples are also provided in "index.html" in the Supplementary Materials for a better understanding.

