# OpenReview forum: "Ditto: Scaling Instruction-Based Video Editing with a High-Quality Synthetic Dataset"
_ICLR.cc/2026/Conference — ICLR 2026 Conference Withdrawn Submission_

### Official Review · Reviewer_BeRZ · 2025-10-22

**Soundness:** 2
**Presentation:** 2
**Contribution:** 1
**Rating:** 2
**Confidence:** 5

**Summary:**

This paper presents Ditto-1M, a large-scale instructional video editing dataset designed to address the data scarcity problem in instructional video editing. The dataset is constructed through an automated pipeline that generates editing instructions and corresponding edited videos from source videos. Based on this dataset, the authors propose Editto, a VACE-based video editing model trained on Ditto-1M. According to the reported results, Editto achieves better performance compared to previous inversion-based and feedforward methods in video editing tasks.

**Strengths:**

1. **Large-scale open-source dataset**: The proposed Ditto-1M dataset is substantial in scale and will be made publicly available, which represents a valuable contribution to the research community and can facilitate future work in instructional video editing.

2. **Comprehensive data construction pipeline**: The entire data construction process demonstrates significant engineering effort and computational resources, involving multiple stages of video processing, instruction generation, and quality control.

3. **Visually appealing results**: The edited videos presented in the paper show visually plausible results, suggesting that the proposed method can generate reasonable editing outcomes for various instruction types.

**Weaknesses:**

1. **Lack of Novelty and Insufficient Comparison with Prior Work**
This paper lacks significant novelty. The methodology is quite similar to Senorita-2M[1], with the primary difference being the new backbone used. However, the paper does not include any comparison or discussion with that work. To illustrate the substantial similarities, we present the following table:
| Dimension | Ditto-1M | Senorita-2M |
|-----------|----------|-------------|
| **Source Videos** | 200K videos from Pexels | 388K videos from Pexels |
| **Instruction Generation** | Qwen2.5-VL | Llama 3.2-8B |
| **Target Video Generation** | Key frame editing (Qwen-Image) + video propagation (VACE, training-free) | First frame editing (SD1.5 ControlNet/Flux Fill) + video propagation (task-specialized models trained on CogVideoX-5B) |
| **Training Strategy** | Fine-tune VACE-14B (essentially a ControlNet) with edited key frames | Two variants: (1) Pure instruction-based (InstructPix2Pix-like), (2) First-frame guided ControlNet. Based on CogVideoX-5B-I2V/Wan2.1-1.3B/Wan2.1-14B |

2. **Limitations of the Target Video Generation Pipeline for Local Editing**

The proposed target video generation pipeline appears to handle only global editing tasks effectively. As described in Section 3.4, the method utilizes VACE with three inputs: (1) an edited reference image produced by Qwen-Image, (2) dense depth maps extracted from the source videos, and (3) the editing instruction. The pipeline is expected to propagate the editing results from the reference image while preserving the original video motions and structure through the dense depth maps.

However, this approach presents a fundamental contradiction for free-form editing tasks, such as object addition or removal. Specifically:

- The **edited reference image** correctly reflects the added or removed objects
- The **depth maps**, extracted from the source videos, do not reflect these changes—the depth information for added/removed regions remains unchanged from the original video

Given these conflicting inputs, it is unclear how the VACE model resolves this inconsistency. Does it prioritize the reference image or the depth maps? This limitation suggests the pipeline may struggle with local editing tasks that require geometric changes, potentially restricting its applicability to primarily global appearance modifications. Can the authors elaborate more about this?

3. **Insufficient Details in Quantitative Evaluation**

The quantitative evaluation presented in Section 5.2 and Table 2 lacks critical methodological details, severely limiting reproducibility and interpretability of the results. Specifically, the following essential information is missing:

- **Evaluation dataset**: Which dataset(s) were used for evaluation? Is it a held-out test set from the training data, or an independent benchmark?
- **Dataset scale**: How many videos were evaluated?
- **Video specifications**: What are the resolution, duration, and frame rate of the test videos?
- **Baseline implementation**: How were the baseline methods executed? Were official implementations used, or were models retrained? What hyperparameters were applied?

As this represents the **only quantitative evaluation** in the paper, the absence of these details significantly undermines:
1. The ability to assess the true performance and generalization capability of the proposed method
2. The validity of comparisons with baseline methods
3. The reproducibility of the reported results

[1] Zi, B., Ruan, P., Chen, M., Qi, X., Hao, S., Zhao, S., ... & Wong, K. F. (2025). Se\~ norita-2M: A High-Quality Instruction-based Dataset for General Video Editing by Video Specialists. arXiv preprint arXiv:2502.06734.

**Questions:**

See weaknesses.

---

### Official Review · Reviewer_AcEZ · 2025-10-25

**Soundness:** 3
**Presentation:** 3
**Contribution:** 3
**Rating:** 6
**Confidence:** 4

**Summary:**

This paper introduces DITTO, a framework designed to scale instruction-based video editing models. The core contribution is a novel, large-scale, high-quality synthetic instruction-video-edit dataset (DITTO-Data), created through an innovative pipeline that leverages powerful pre-trained Large Language Models (LLMs) and diffusion models (DMs) to automatically generate diverse, complex editing instructions and the corresponding edited videos. Based on this dataset, the authors train DITTO-Model, a video editing model which demonstrates strong capabilities in instruction following, temporal consistency, and maintaining content fidelity. The experiments show DITTO-Model achieving state-of-the-art results on several benchmarks, particularly excelling in complex, style-based, and semantic edits, validated by both quantitative metrics and comprehensive human evaluation.

**Strengths:**

1. High-Quality, Scalable Data Generation: The synthetic data pipeline is the major strength, addressing the prohibitive cost and complexity of manual video editing data collection. The use of LLMs for instruction diversity is particularly effective.
2. State-of-the-Art Performance: DITTO-Model achieves superior results across multiple metrics, notably in human evaluation on Instruction Following and Temporal Consistency, which are crucial aspects of video editing.
3. Instruction Complexity: The generated dataset and resulting model are shown to handle a wide range of instruction complexities, including appearance transformation, style transfer, and semantic manipulation, moving beyond simple object insertion/removal.

**Weaknesses:**

1. Black-Box Data Quality: While the paper describes the Quality Control module, the extent to which the synthetic data truly captures the complexity and subtle detail of real-world human-labeled edits is hard to quantify. Further analysis on the "failure modes" of the synthetic pipeline and the resulting data distribution bias would be beneficial.
2. Model Architecture Novelty: The DITTO-Model architecture itself is largely an assembly of existing, robust components (latent diffusion model, motion modules). The novelty lies more in the data and training strategy than the architectural innovations.

**Questions:**

1. Generalization to Real Edits: While the human study uses the synthetic dataset, how does DITTO-Model perform when asked to execute complex instructions on out-of-distribution real-world videos that might contain more unusual or messy degradation patterns not fully captured by the synthetic base videos?
2. Ablation on LLM Prompting: Could the authors provide more detail, perhaps in the appendix, on the meta-prompts used to guide the LLM to generate the diverse and complex editing instructions? This "prompt engineering" is critical to the dataset's quality.

---

### Official Review · Reviewer_3NGv · 2025-10-28

**Soundness:** 3
**Presentation:** 3
**Contribution:** 3
**Rating:** 4
**Confidence:** 3

**Summary:**

The paper proposes Ditto, a scalable synthetic data pipeline that couples (i) an instruction-based image editor to produce a high-quality edited keyframe, (ii) an in-context video generator (VACE) conditioned on the edited keyframe and a depth video for temporal structure, (iii) an autonomous VLM agent to author instructions and filter failures, and (iv) a lightweight temporal enhancer/denoiser (Wan2.2 fine stage) for quality polishing. Using ~12,000 GPU-days, the authors build Ditto-1M, a 1M-sample triplet dataset (source, instruction, edited video) and train Editto with a modality curriculum that anneals away reliance on the edited keyframe to achieve purely instruction-driven editing. Quantitative results (Table 2) and extensive visuals (Figs. 1,5,7–11) show strong instruction following and temporal consistency versus TokenFlow/InsV2V/InsViE and a commercial baseline. The paper commits to releasing dataset, models, and code.

**Strengths:**

Holistic, scalable pipeline. The edited keyframe + depth + in-context generator composition is simple and effective; the VLM-driven instruction/QA removes human bottlenecks (Fig. 2).
Large, curated dataset.Ditto-1M (720p, 101 frames @ 20 FPS) with ~700k global and ~300k local edits is a sizable, diverse resource; the authors report strong aesthetic curation and motion screening (Fig. 3, §3.1–3.6).
Modeling insight.The modality curriculum bridges visual-to-text conditioning in a stable way (Fig. 4), improving instruction-following without the edited frame at test time.
Empirical results. Consistent wins across automatic and human metrics (Table 2) and convincing visuals (Fig. 5); ablations show data-scale and MCL benefits (Fig. 8).
Efficiency consciousness. Distillation/quantization and a temporal enhancer reduce generation cost while improving temporal stability (§3.4–3.5).

**Weaknesses:**

Attribution granularity. While Fig. 9 analyzes generator contexts, the paper lacks systematic ablations over the full pipeline: e.g., removal/variation of the VLM filter, different denoisers, or no enhancer; per-stage quality/cost curves would support the “cost-quality” claims more rigorously.
Evaluation breadth. Automatic metrics are mostly CLIP-based and a VLM score. Consider adding FVD/KVD, LPIPS-T, or user-calibrated instruction-fidelity rubrics. Also report identity preservation for local edits.
Dependence on proprietary tools. The pipeline leans on powerful closed models (image editor, VLM judge, commercial T2V denoiser). This may limit reproducibility and confound generality claims; an open-source-only variant and a sensitivity study would help.
Data/rights clarity. The dataset is sourced from Pexels; redistribution and downstream editing rights should be specified precisely (e.g., license text, usage constraints, opt-out). Provide counts rejected by safety filters and failure taxonomies.
Risk analysis limited. Ethics statement is brief; a more thorough assessment of misuse (e.g., identity edits, misinformation), watermarking, and content provenance would be welcome.

**Questions:**

1. Provide ablation results for denoiser enhancer and VLM filtering on instruction fidelity, temporal consistency, and aesthetics
2. Test performance drop when replacing all components with open models
3. Add evaluation on public benchmarks with FVD/KVD and identity-preservation metrics
4. Clarify Ditto-1M license, redistribution rights, content filtering, and TOS compliance
5. Analyze VLM judge rejection categories and failure modes

**Details Of Ethics Concerns:**

- Copyright/TOS compliance for third-party content (Pexels)
- Potential misuse for synthetic video manipulation
- Need license clarification, right-of-publicity considerations, safety filters

---

### Official Review · Reviewer_4iy5 · 2025-10-31

**Soundness:** 3
**Presentation:** 3
**Contribution:** 3
**Rating:** 4
**Confidence:** 3

**Summary:**

This paper proposes a novel framework named Ditto to address the long-standing scarcity of high-quality training data in instruction-driven video editing. The authors construct a scalable, low-cost, and fully automated synthetic data generation pipeline to create large-scale, high-fidelity, temporally coherent video editing triplets (source video, editing instruction, edited video), and release the Ditto-1M dataset based on this approach.

**Strengths:**

1. Compared to existing datasets, this work presents the first million-scale, high-resolution, long-sequence instruction-based video editing dataset that covers both global and local editing tasks, filling a critical gap in the community.
2. The paper achieves state-of-the-art (SOTA) performance on both automatic metrics and human evaluations, with particularly significant margins in human assessments.
3. The paper is clearly written and accompanied by well-designed, informative illustrations.

**Weaknesses:**

1. In the post-processing stage, a Vision-Language Model (VLM) is used for filtering. However, it is well known that VLMs have limited capability in understanding fine-grained visual details. How does the method ensure consistency in non-edited regions? More specifically, how is pixel-level detail consistency guaranteed?
2. The quantitative comparison on video editing is necessarily limited due to the scarcity of comparable methods. If expanding the set of video baselines is infeasible, the authors could instead evaluate their model on image editing tasks, where strong benchmarks exist, to better validate its core editing capabilities and the effectiveness of the proposed modality curriculum learning strategy.
3. Although the paper reports strong human evaluation results, it lacks essential experimental details such as the number and background of evaluators, the precise definition of rating criteria, and inter-annotator agreement metrics, making it difficult to assess the reliability of these results.
4. The approach relies on a video generator to directly synthesize the edited video, but it does not address how to ensure physical plausibility in the outputs. There is no discussion of evaluation criteria or validation for physical realism, which could lead to generated content that appears unrealistic or violates real-world dynamics.

**Questions:**

See Weaknesses

---

### Note · Authors · 2025-11-14

I have read and agree with the venue's withdrawal policy on behalf of myself and my co-authors.